# BOOST: Harnessing Black-Box Control to Boost Commonsense in LMs' Generation

## Yufei Tian  Felix Zhang  Nanyun Peng

Department of Computer Science, University of California, Los Angeles

{yufeit,violetpeng}@cs.ucla.edu

## Abstract

Large language models (LLMs) such as GPT-3 have demonstrated a strong capability to generate coherent and contextually relevant text. However, amidst their successes, a crucial issue persists: their generated outputs still lack commonsense at times. Yet fine-tuning the entire LLM towards more commonsensical outputs is computationally expensive if not infeasible. In this paper, we present a computation-efficient framework that steers a frozen Pre-Trained Language Model (PTLM) towards more commonsensical generation (i.e., producing a meaningful and plausible output that incorporates a list of concepts).

Specifically, we first construct a reference-free evaluator that assigns a sentence with a commonsensical score by grounding the sentence to a dynamic commonsense knowledge base from four different relational aspects. We then use the scorer as the oracle for commonsense knowledge, and extend the controllable generation method called NADO to train an auxiliary head that guides a fixed PTLM to better satisfy the oracle. We test our framework on a series of `GPT-2`-, `FLAN-T5`- and `Alpaca`-based language models (LMs) on two constrained concept-to-sentence benchmarks. Human evaluation results demonstrate that our method consistently leads to the most commonsensical outputs.[1]

## 1 Introduction

Recent years have witnessed remarkable progress in massively Pre-Trained Language Models such as GPT-3 (Brown et al., 2020), Llama (Touvron et al., 2023) and instruction following models such as Flan-T5 (Chung et al., 2022), ChatGPT (OpenAI, 2022), and Alpaca (Taori et al., 2023). However, one significant drawback is the lack of commonsense knowledge in their generated texts. There have been criticisms around their commonsense

---

[1]Source code will be available at https://github.com/PlusLabNLP/BOOST_EMNLP23

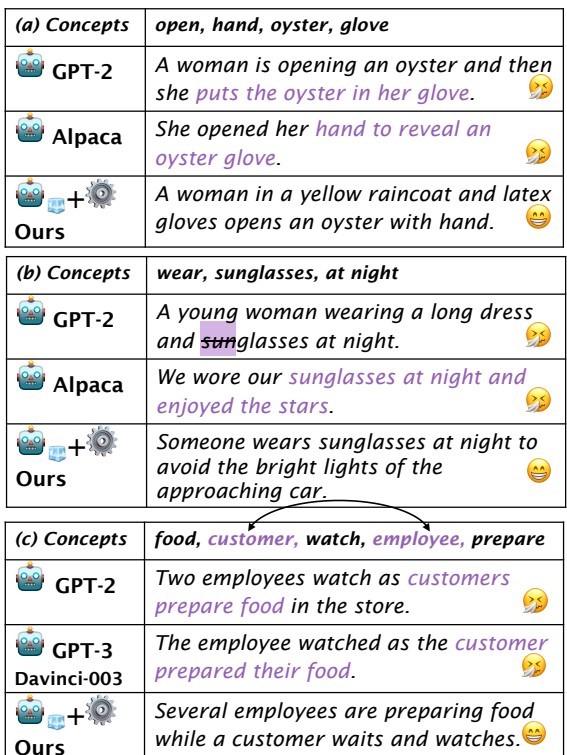

| (a) Concepts | open, hand, oyster, glove |
| --- | --- |
| GPT-2 | *A woman is opening an oyster and then she puts the oyster in her glove.* 🤢 |
| Alpaca | *She opened her hand to reveal an oyster glove.* 🤢 |
| Ours | *A woman in a yellow raincoat and latex gloves opens an oyster with hand.* 😄 |

| (b) Concepts | wear, sunglasses, at night |
| --- | --- |
| GPT-2 | *A young woman wearing a long dress and ~~sun~~glasses at night.* 🤢 |
| Alpaca | *We wore our sunglasses at night and enjoyed the stars.* 🤢 |
| Ours | *Someone wears sunglasses at night to avoid the bright lights of the approaching car.* 😄 |

| (c) Concepts | food, *customer*, watch, *employee*, prepare |
| --- | --- |
| GPT-2 | *Two employees watch as customers prepare food in the store.* 🤢 |
| GPT-3 Davinci-003 | *The employee watched as the customer prepared their food.* 🤢 |
| Ours | *Several employees are preparing food while a customer waits and watches.* 😄 |

Figure 1: LMs such as `GPT-2` finetuned, `Alpaca-7b` fewshot, and `GPT-3 Davinci-003` fail to incorporate the concepts in a commonsensical way. We highlight the insensible phrases in purple. (c) illustrates that they are also vulnerable to perturbations of the input prompt as simple as the swap of two concept positions. Our system which uses an auxiliary model to steer a **frozen** PTLM generates the most commonsensical outputs.

impotence (Marcus, 2020; Elazar et al., 2021; Mahowald et al., 2023), and a discrepancy in what LLMs generate in the wild versus in question answering (Chen et al., 2023).

In this paper, we explore the task of generative commonsense reasoning: a constrained text generation task aiming to generate a plausible sentence given a list of concepts as input. As depicted in Figure 1, language models should generate a sentence that incorporates 'open, hand, oyster, glove' in a meaningful way that aligns with our commonsense. We unveil that *LLMs are*

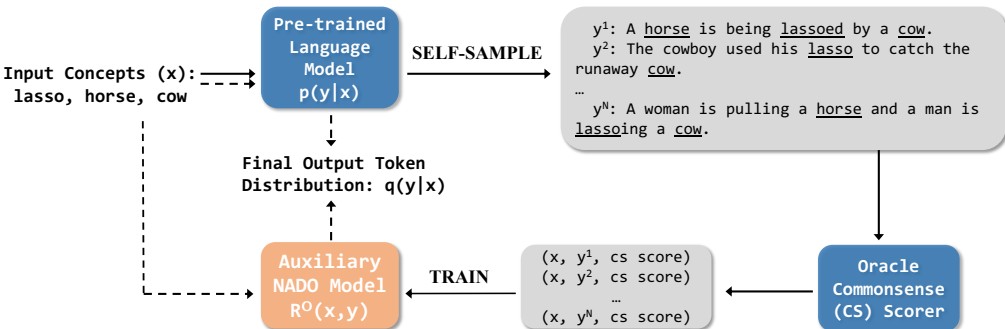

Figure 2: The process of BOOST to steer a frozen PTLM with an additional neural model and oracle commonsense scorer. The solid lines indicate the training process, while the dashed lines indicate inference. In practice, we combine our commonsense scorer with lexical checking rules, and use the joint signal to train the auxiliary model.

*unreliable and fail to generate commonsensical outputs* when the input concepts get complicated. In another case depicted in Figure 1(c), when we swap the position of two input concepts 'customer' and 'employee', LLMs such as Davinci-003 are vulnerable to the change and generate *'employee watched a customer prepare food'* despite being instructed to not consider the concept appearance order, which is far from plausible.

Various knowledge-augmented systems have been previously proposed to incorporate external knowledge into the model (Liu et al., 2021; He et al., 2022) for more plausible generation outputs. However, they all require updating model weights at the scale of hundreds millions of parameters such as BART (Lewis et al., 2020). As PTLMs continue to evolve and scale up to hundreds of billions of parameters in size, finetuning the entire LM becomes computationally prohibited for many parties in academia and the industry.

In this work, we propose BOOST, a framework to boost the commonsense of PLTMs' generation in a plug-and-play manner (Figure 2), which is inspired by the recent development of controllable generation to use a small auxiliary model to control a PTLM by training on its *self-generated* samples (Meng et al., 2022). Specifically, to better integrate commonsense knowledge, we first build a scorer that evaluates how commonsensical a sentence is. The commonsense scorer, called $\mathcal{O}$-Scorer, extracts tuples of commonsense-related concepts (e.g., <customers, prepare their food>) from a sentence, and scores the extracted tuples by grounding the tuples to a dynamic commonsense knowledge base (CSKB) (Bosselut et al., 2019; Ghazarian et al., 2023). Next, we use the signal from the $\mathcal{O}$-Scorer to train an auxiliary model that steers the PTLM toward more commonsensical outputs.

Note that our training process is generalizable and only requires access to the output probability of the PTLMs, which is also efficient due to the smaller size of the auxiliary model.

We test our method on gpt-2, Alpaca, and Flan-T5 on two datasets: 1) CommonGen (Lin et al., 2020) that focuses on daily concepts (e.g., <open, hand, oyster, glove>) and 2) CSK-PN (Chen et al., 2023) that contains concepts linked with negated commonsense relations (e.g. <wear sunglasses, at night>).

Our contributions are two-fold. First, we propose a reference-free evaluator to assess how commonsensical a sentence is, which achieves on-par performance with referenced-based metrics such as BERTSCore (Zhang et al., 2019) in terms of correlation with human judgment. Second, we extend a controllable generation approach to improve commonsense for black-box PTLM. Experimental results show that our method consistently results in the most commonsensical outputs.

## 2 Methodology

### 2.1 Overview

Figure 2 provides an overview of our approach, BOOST. During training, BOOST first generate numerous samples $(\mathbf{y}^1, ..., \mathbf{y}^N)$ from the PTLM conditioned on the input constraint $\mathbf{x}$ (e.g., 'lasso horse cow'). We then construct an oracle to give commonsense scores on all of these self-sampled generations. Next, for each $\mathbf{y}^i$ of length $T_i$, we train the auxiliary model called NADO which essentially learns to predict the expected cs score of the complete sequence $\mathbf{y}^i$ given $\mathbf{x}$ and an incomplete sequence $\mathbf{y^i}_{<t}$ ($t \in [1, 2, ..., T_i]$). The flow at inference time is illustrated in dashed lines: both the PTLM and NADO take $\mathbf{x}$ and the generated sequence (prefix) $\mathbf{y}_{<L}$ as input, from which we

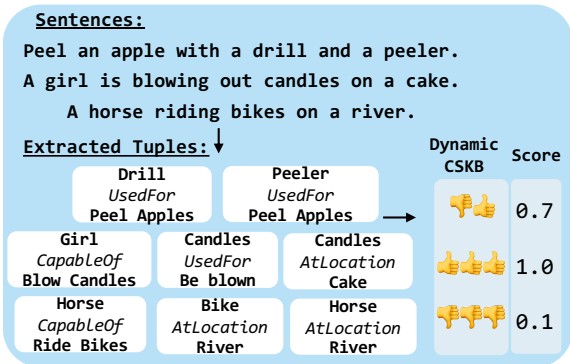

Figure 3: An example of our oracle commonsense scorer. We first extract tuples from a target sentence, and assign each extracted tuple with a commonsensical score using COMET (Bosselut et al., 2019), a dynamic commonsense knowledge base. The sentence-level score is then obtained by aggregating tuple-level scores.

obtain the final output distribution $q(\mathbf{y}|\mathbf{x})$.

The rest of this section is organized as follows. In §2.2, we first introduce details to construct the commonsense scorer. Then, in §2.3, we provide the theory and practices to train the auxiliary model on PTLM's self-generated data towards the oracle.

## 2.2 Constructing Commonsense Scorer

We use commonsense relation tuples as the intermediate representation of a sentence. Specifically, we get rid of human annotation and leverage on the results of few-shot LLMs. We then check whether these extracted tuples are sensible. To this end, we assign each parsed tuple with a compatibility score based on its maximum similarity with the numerous valid accepted answers generated by COMET, a dynamic commonsense knowledge base (CSKB). Scores for all tuples in a target sentence are then aggregated to obtain the sentence-level commonsense score. Figure 3 provides an illustration of our oracle scorer.

### 2.2.1 Commonsense-Relation Extraction

**Tuple Format** We leverage the format of ConceptNet (Speer et al., 2017), a widely used knowledge graph connecting concepts or events with commonsense relations to represent general world knowledge. Specifically, each tuple $\mathcal{T}$ contains a head concept/event $h$ (e.g., *driller*) and a tail concept/event $t$ (e.g., *drill a hole*), which are connected through a commonsensical relation $r$ (e.g., *is Used For*). We consider four crucial relation types that dominantly exist: *is UsedFor*, *is Capable Of*, *is At Location*, and *is Part Of*.

**Tuple Extraction** We present a labor and cost efficient way to extract all tuples from a target sentence, including both commonsensical and nonsensical tuples. LLMs such as GPT-3 and ChatGPT (Brown et al., 2020; Ouyang et al., 2022) have demonstrated remarkable ability of few-shot in-context learning for semantic parsing tasks (Dong and Lapata, 2016; Dunn et al., 2022). Motivated by such progress, instead of asking human workers to annotate a training set of sentences, we leverage OpenAI's GPT3.5-Turbo model to parse the relevant tuples. We hand-crafted 9 examples for our few-shot prompt such that the LLM can accurately extract both sensical tuples (e.g., a girl *is Capable Of* blowing candles) and nonsensical tuples (e.g., horse *is Capable Of* riding bikes) from the input sentence. The complete instruction and prompt can be found in Appendix A.

However, in practice, using GPT-3.5-Turbo to parse *all* sentences needed to train our auxiliary model is costly and unreliable when dependent on the unpredictable traffic of OpenAI's API. To obtain an extractor that can parse $\sim$ a million sentences at a reasonable cost, we finetune a T5 large model (Raffel et al., 2020) on 6,000 GPT-3.5 annotated sentences for the same task. We show the performance of both tuple extractors in §3.2.

### 2.2.2 Generative Commonsense Scoring

After extracting relation tuples from a sentence, we need to assess how commonsensical they are. To this end, we follow the compatibility test proposed by Ghazarian et al. (2023) and leverage COMET (Bosselut et al., 2019), a pre-trained generative commonsense transformer that can predict sensible tails given the heads and relations as input. Compared to other fixed and predefined knowledge bases, COMET is dynamic and much more flexible when dealing with original and unseen inputs.

Formally, given a tuple $\mathcal{T}_i = (h_i, r_i, t_i)$ and a dynamic CSKB denoted by $\mathcal{C}_{dy}$, we query $\mathcal{C}_{dy}$ with the head $h$ and relation $r$ to obtain a diverse list of conditionally generated tails with beam decoding: $\{t_j^*\}_{j=1}^k = \mathcal{C}_{dy}(h_i, r_i, beam = k)$. The commonsense score for $\mathcal{T}$ is computed by

$$\text{COMPAT}(\mathcal{T}_i|\mathcal{C}_{dy}) = \max_{1 \leq i \leq k} \cos(\text{emb}(t_i), \text{emb}(t_j^*)),$$
(1)

where $emb(\cdot)$ is the vector representation from a sentence embedding model (Reimers and Gurevych, 2019). Finally, we need to aggregate the compatibility scores computed from different

triplets extracted from a single sentence. The sentence-level commonsense score is denoted as the $\mathcal{O}$-score. One rationale is that a single non-sensical tuple can result in a nonsensical sentence, while the other is that one mistake will be mitigated by other reasonable tuples. We hence take the 1) minimum and 2) average compatibility scores, and study their correlation with human judgement in §3.3 and Table 2.

## 2.3 Commonsense-Guided Generation

In this subsection, we describe how we use our derived commonsense oracle to steer the PTL) toward more commonsensical outputs through a neurally-decomposed head (NADO). In §2.3.1, we summarize the theoretical solution of Meng et al. (2022) to decompose the sequence-level oracle into token-level guidance with a ***frozen*** PTLM, such that when generating the $i$-th token, the auxiliary neural network modifies the original output logits predicted by the PTLM. Then, in §2.3.2, we leverage this method to generate more commonsensical outputs. Note that our model only trains the *additional* NADO head which has much smaller size than the PTLM and does not require access to the parameters inside the PTLM.

### 2.3.1 Token-Level Guidance with NADO

**Notation** Suppose we have a sub-optimal PTLM $p(\mathbf{y}_{t=T'}|\mathbf{x}, \mathbf{y}_{t<T'})$, our goal is to obtain an optimal auto-regressive model $q$ from $p$ such that $q$ generates outputs better satisfying the oracle scorer $\mathcal{O}$ (for example, $q$'s generated outputs achieve higher $\mathcal{O}$-scores than $p$). We now define a **predictive function** $R^{\mathcal{O}}(\mathbf{x}, \mathbf{y}_{t<T'})$ that predicts the *expected $\mathcal{O}$-scores* of the complete sequence $\mathbf{y}$ given input $\mathbf{x}$ and the currently generated tokens $\mathbf{y}_{t<T'}$.

$$R^{\mathcal{O}}(\mathbf{x}, \mathbf{y}_{t<T'}) = \mathrm{Exp}_{\mathbf{y} \sim p(\mathbf{y}|\mathbf{x})}\left[\mathcal{O}(\mathbf{x}, \mathbf{y}) \mid \mathbf{y}_{<T'}\right]$$
(2)

$$= \sum_{\mathbf{y} \in \mathcal{Y}} p(\mathbf{y} \mid \mathbf{x}, \mathbf{y}_{t<T'})\, \mathcal{O}(\mathbf{x}, \mathbf{y})$$
(3)

**Solution** The unique closed formed solution of the optimal $q$ is (namely, generates most commonsensically according to $\mathcal{O}$):

$$q^{*}(y_{=T'} \mid \mathbf{x}, \mathbf{y}_{<T'}) = \frac{R^{\mathcal{O}}(\mathbf{x}, \mathbf{y}_{\leq T'})}{R^{\mathcal{O}}(\mathbf{x}, \mathbf{y}_{\leq T'-1})} p(y_{=T'} \mid \mathbf{x}, \mathbf{y}_{<T'})$$
(4)

Please refer to Meng et al. (2022) for details of the proof. From Eq.4 we see that when both $\mathbf{x}$ and $\mathbf{y}_{t<T'}$ are fixed, the optimal auto-regressive model is factorized into $R^{\mathcal{O}}$ and $p$ at step $T'$:

$$q^{*}(y_{T'} \mid \mathbf{x}, \mathbf{y}_{<T'}) \propto R^{\mathcal{O}}(\mathbf{x}, \mathbf{y}_{\leq T'}) \cdot p(y_{T'} \mid \mathbf{x}, \mathbf{y}_{<T'})\ \text{(5)}$$

**Approximation** As we cannot enumerate $\mathcal{Y}$ that contains infinite number of sequences, the well-defined $R^{\mathcal{O}}$ is intractable. A neural model called NADO is hence introduced to approximate $R^{\mathcal{O}}$, by training on numerous samples $\mathcal{Y}$ generated by $p$.

### 2.3.2 NADO-Guided Generation

Given a pre-trained language model $p$ such as the GPT-2 and Alpaca model, we first ask $p$ to generate numerous samples to obtain an approximation of $\mathcal{Y}$ with various inputs concepts $\mathbf{x} \in \mathcal{X}$. We then use the oracle $\mathcal{O}$ to assign each sample a score, which is used to train the NADO model.

**Training** During training, the NADO model takes $\mathbf{x}, \mathbf{y}$ as input, and learns to predict from $R^{\mathcal{O}}(\mathbf{x}, \mathbf{y}_{t=0})$ till $R^{\mathcal{O}}(\mathbf{x}, \mathbf{y}_{t\leq T})$. Here, $T$ is the complete sequence length and the sentence-level value $\mathcal{O}(\mathbf{x}, \mathbf{y})$ is used as the labels for all steps, from $t=0$ till $t=T$. We emphasize that in order for $\mathcal{O}$ to learn $R^{\mathcal{O}}$ successfully, all $(\mathbf{x}, \mathbf{y})$ pairs must be self-sampled by the base model $p$ instead of come from the CommonGen training data.

We use cross entropy loss as the objective function. Given a particular input $\mathbf{x}$, the cross entropy loss is

$$\mathcal{L}_{CE}(\mathbf{x}) = \sum_{\mathbf{y} \in \mathcal{Y}} p(\mathbf{y} \mid \mathbf{x}) L_{CE}(\mathbf{x}, \mathbf{y}, R^{\mathcal{O}})$$
$$= \sum_{i=0}^{T} CE\left(R^{\mathcal{O}}(\mathbf{x}, \mathbf{y}_{\leq i}), \mathcal{O}(\mathbf{x}, \mathbf{y}_{\leq i})\right)$$
(6)

In practice, we also add a regularization term to the loss. In order to satisfy the definition that $\sum_{y_i} R^{\mathcal{O}}(\mathbf{x}, \mathbf{y}_{\leq i})\, p(y_i \mid \mathbf{x}, \mathbf{y}_{<i}) = R^{\mathcal{O}}(\mathbf{x}, \mathbf{y}_{\leq i-1})$, our regularization loss is measured by the KL divergence of the following:

$$\mathcal{L}_{reg}(\mathbf{x}) = KL(\sum_{y_i} R^{\mathcal{O}}(\mathbf{x}, \mathbf{y}_{\leq i}) \cdot p(y_i \mid \mathbf{x}, \mathbf{y}_{<i}),$$
(7)
$$R^{\mathcal{O}}(\mathbf{x}, \mathbf{y}_{\leq i-1}))$$
(8)

Then, the final training loss is $\mathcal{L}_{CE}(\mathbf{x}) + \lambda \mathcal{L}_{reg}(\mathbf{x})$, where $\lambda$ is a hyper-parameter to balance these two

terms. In practice, we use grid search and choose the best $\lambda$ from [0.1, 0.5, 1.0].

**Inference**   At inference time, there are two forward passes as shown in Eq.5 and Figure 2. The decoding efficiency roughly remains unchanged because the NADO head has much smaller size than the base PTLM.

## 3   Experimental Results for the Oracle

In this section, we show the results of the commonsense scorer described in §2.2. The experiments and results of commonsense-guided generation (§2.3) can be found in §4 and §5.

### 3.1   Tuple Extraction Data

We use the GPT-3.5-Turbo model provided by OpenAI to extract the tuples of 6,000 sentences (with a total cost of $12.4), based on which we train the T5-large based tuple extractor. Since our goal is to parse all possible commonsense tuples whether they are sensical or not, we need both sensical and less reasonable sentences. To this end, we randomly select 3,000 sentences from the CommonGen (Lin et al., 2020) train split (we consider them as more sensical) and another 3,000 sampled from a basic gpt-2 model (we consider them as less coherent and sensical).

### 3.2   Tuple Extractor Results

Following the rationale in §3.1, we study the benefit brought by augmenting the training data with tuples extracted from less coherent and sensical sentences. Specifically, we compare the following three settings: 1) base: trained on the 3,000 sensical sentences; 2) aug: trained on 1,500 sensical sentences and 1,500 less sensical sentences; 3) all: trained on all 6,000 sentences. We test the model performance on a held-out set of 350 sentences that is mix of both types. To obtain the gold labels on the test set, we start with the few-shot GPT-3.5's annotation. After that, two human annotators iteratively checked and fixed any error they see.

For each relation type, we report the average f1-score in Table 1. Here, if the lemmatized tokens in a generated triplet has over 50% overlap with those in the ground-truth triplet, we consider it as correct. Otherwise, we consider it as wrong. Comparing T5-Large aug with T5-Large base in Table 1, we see improvements across all four relation types. Besides, increasing the train data size also boosts the extractor's performance. We also notice that

| Relation Type | At Lo-cation | Used For | Capable Of | Part Of | All |
|---|---|---|---|---|---|
| T5-Large base | 71.0 | 67.1 | 65.6 | 76.8 | 70.1 |
| T5-Large aug | 72.4 | 67.9 | 68.9 | 79.2 | 72.1 |
| T5-Large all | 73.4 | 70.5 | 69.4 | 79.6 | **73.2** |
| Few-Shot GPT-3.5 | 83.5 | 71.1 | 78.7 | 82.2 | 78.9 |

Table 1: The performance of different tuple extractors, measured by F1-score. The last row indicates the upper bound that our T5 models can achieve.

| Reference-Free: min \| mean | | Reference-Based | |
|---|---|---|---|
| T5 $\mathcal{O}$-Score | 0.276 \| 0.284 | METEOR-all | 0.214 |
| GPT-3.5 $\mathcal{O}$-Score | 0.281 \| 0.299 | BERTScore-one | 0.280 |
| Gold $\mathcal{O}$-Score | **0.346 \| 0.365** | BERTScore-all | 0.302 |

Table 2: Spearman correlation between human commonsense ratings and six automatic metrics: our $\mathcal{O}$-Score with tuples extracted by T5, GPT-3.5-Turbo, and the gold tuples, plus METEOR and BERTScore.

our extractors perform worse on *UsedFor* and *CapableOf* than on *AtLocation* and *PartOf*, which is partially due to the errors of the training signal (i.e., labels are inaccurately annotated by GPT-3.5).

### 3.3   Oracle Commonsense Scorer Results

To compute the machine-generated compatibility score in Eq.1, we set beam size $k = 128$. Meanwhile, we instruct human annotators to evaluate the target sentences on how commonsensical they are. Each sentence is annotated by 3 workers with a scale of 1 (least) to 4 (best). We also ask every annotator to specify which part of the target sentence is nonsensical. We find out that explicitly asking the workers to pay detailed attention and point out the erroneous parts helps to increase the inter annotator agreement (IAA, measured by Spearman's correlation) from 0.56 to 0.67. The final sentence-level commonsense score annotated by humans is the average of 3 individual ratings.

Table 2 shows the correlations between human ratings and automatic scores. For our proposed $\mathcal{O}$-Score, we report the correlations of taking the minimum (min) and average (mean) of all tuple-level compatibility scores. Taking the average consistently result in higher correlation, reflecting that one mistake of a nonsensical tuple can be mitigated by other sensical ones. Therefore, we use the mean score to train the auxiliary model. We also compare with reference-based metrics such as METEOR (Banerjee and Lavie, 2005) and BERTScore(Zhang et al., 2019). Since there are, on average, 4 references per candidate in the Com-

monGen dataset, we select the first reference to compute BERTScore-one, and all available references to compute BERTScore-all. We show that our reference-free scorer performs on par with the best reference-based metric, BERTScore-all, and outperforms the same when use gold tuples extracted by human.

## 4 Experiments about Guided Generation

### 4.1 Data

**Training Data** As is illustrated in Figure 2, we train our auxiliary model on the PTLM's self-sampled data. For each set of input concept, we use top-p sampling (p=0.95) with temperature T=0.7 to generate $N$ samples. In theory, the larger the $N$, the more accurate approximation $R^{\mathcal{O}}$ can learn. In practice, due to limitations in computational resources, we set $N$ to 48 when the base model $p$ is `gpt-2`, and 10 for `Alpaca`. In total, we have 1.5M training instances self-sampled by `gpt-2` and 0.3M training instances self-sampled by `Alpaca`.

**Test Data** We test on two different datasets. The first is the CommonGen dev split (Lin et al., 2020) which contains 993 lists of keywords focusing on daily concepts (e.g., *<open, hand, oyster, glove>*). Each list of keywords is paired with more than one human written references. Our second test data is distilled from `CSK-PN` (Chen et al., 2023), which sources challenging triples from Concept-Net (Speer et al., 2017) and tags them with positive/negative relation labels. We randomly select 993 triples with negative relations from `CSK-PN` (e.g. *<wear sunglasses, at night>*). There is no human reference for the second set. To reduce the effect of data leakage in GPT-3 and Alpaca, we randomly shuffled the keywords within each entry.[2]

### 4.2 Experimental Setup

**Choice of Base Models.** Although our framework does not require fine-tune PTLMs, it does require access to the PTLM's *output distribution*. Hence, we cannot apply our method to some popular but close-sourced LLMs such as ChatGPT. We choose `Alpaca`, `Flan-T5`, and `gpt2` instead. In addition, because the pre-trained `gpt2` has no instruction following abilities, we have to train it to learn the task of 'generating a commonsensical sentence given these input concepts'. Specifically, we finetune

it on the CommonGen training data for 1 epoch, well before the finetuning converges. We call this process *warm up*, as the goal is mainly to get the smaller base model onboard with our task format. For instruction-following models such as `Alpaca`, we still add this warm up process for a fair comparison. In total, we apply our commonsense-guided generation method to 5 different base models: `gpt-2-large` with warm up, zero-shot `Alpaca-7b`, few-shot `Alpaca-7b`, `Alpaca-7b` with warm up, and zero-shot `Flan-T5-large`.

**Auxiliary Models.** The auxiliary $R^{\mathcal{O}}$ models are 4-layer transformer decoders with the same dimension and number of heads as the base models. [3] They are $1/9$, $1/8$, and $1/12$ the size of `gpt-2-large`, `Alpaca-7b`, and `Flan-T5-large`. We train the auxiliary models for 10 epochs with a learning rate of $1e-5$ on a single NVIDIA A100 80GB GPU. In comparison, it is not possible to finetune `Alpaca-7b` using only one 80GB GPU without any memory-saving technique such as LoRA (Hu et al., 2021).

### 4.3 Compared Systems

**A*esque Decoding (Lu et al., 2022)** A Neurologic decoding algorithm that injects constraints into a neurologic process with a look ahead heuristic, which results in more plausible outputs.

**Gelato (Zhang et al., 2023)** A tractable probabilistic model (TPM) to impose constraints in language models such as `gpt2-large`. It achieves state-of-the-art (SOTA) performance on constraint satisfaction. Because it is non-trivial to train new TPMs on `Alpaca`-based models, we use the authors' original TPM which is trained on the `gpt2-large` model that is finetuned on CommonGen.

**Lex (Meng et al., 2022)** The vanilla NADO method trained only with lexical constraints as the sequence-level Boolean oracle. Namely, the scorer returns 1 if all lexical constraints are satisfied, and 0 otherwise.

**BOOST (Ours)** Our method that uses the commonsense oracle to steer the auxiliary NADO model. We compare two variations: **1)** BOOST CS: using only the commonsense oracle introduced in §2.2, **2)** BOOST Joint: multiplying the lexical checking Boolean function (the same used in **Lex**) with the commonsense oracle score.

---

| Test Data | CommonGen (Lin et al., 2020) | | | | | CSK-PN (Chen et al., 2023) | | | |
|---|---|---|---|---|---|---|---|---|---|
| | Automatic | | | Human | | Automatic | | Human | |
| Evaluation Metric | $\mathcal{O}$ Score | Coverage | BLEU4 | CS | Overall | $\mathcal{O}$ Score | Coverage | CS | Overall |
| *Setting:* gpt2 *warm up* | | | | | | | | | |
| A*esque (Lu et al., 2022) | 0.469 | 97.2% | 28.1 | 2.37 | 2.72 | 0.489 | 63.0% | 3.14 | 3.09 |
| GeLaTo (Zhang et al., 2023) | 0.592 | 99.3% | 33.0 | 2.45 | 2.78 | / | / | / | / |
| Base Model | 0.514 | 90.7% | 23.2 | 2.31 | 2.80 | 0.53 | 83.9% | 3.10 | 3.04 |
| Lex (Meng et al., 2022) | 0.538 | 96.1% | 29.8 | 2.38 | 2.80 | 0.544 | 92.1% | 3.14 | 3.06 |
| BOOST CS | 0.615 | 90.9% | 23.6 | **2.64** | **3.12** | 0.595 | 89.2% | **3.33** | 3.13 |
| BOOST Joint | 0.597 | 96.1% | 30.1 | 2.54 | 3.01 | 0.587 | 92.0% | 3.28 | **3.18** |
| *Setting:* Flan-T5 *zero-shot* | | | | | | | | | |
| Base Model | 0.571 | 84.6% | 17.5 | 2.86 | 2.80 | 0.555 | 80.7% | 2.78 | 2.71 |
| Lex | 0.577 | 93.7% | 26.0 | 3.04 | 2.92 | 0.569 | 89.6% | 2.97 | 2.88 |
| BOOST CS | 0.619 | 91.3% | 21.6 | **3.14** | 3.05 | 0.613 | 88.9% | 3.07 | **3.03** |
| BOOST Joint | 0.606 | 93.1% | 25.6 | 3.12 | **3.06** | 0.601 | 89.6% | **3.08** | 3.00 |
| *Setting:* Alpaca *warm up* | | | | | | | | | |
| Base Model | 0.563 | 91.5% | 20.9 | 3.02 | 2.84 | 0.523 | 93.9% | 3.07 | 3.06 |
| Lex | 0.584 | 95.9% | 30.5 | 3.12 | 3.00 | 0.535 | 95.0% | 3.14 | 3.10 |
| BOOST CS | 0.611 | 93.6% | 28.9 | **3.36** | **3.11** | 0.558 | 94.4% | 3.21 | 3.19 |
| BOOST Joint | 0.592 | 95.7% | 30.3 | 3.32 | **3.11** | 0.543 | 94.8% | **3.23** | **3.21** |
| *Setting:* Alpaca *zero-shot* | | | | | | | | | |
| Base Model | 0.509 | 90.4% | 21.3 | 2.98 | 3.07 | 0.536 | 93.2% | 3.26 | 3.09 |
| Lex | 0.566 | 95.3% | 30.1 | 3.03 | 3.05 | 0.547 | 95.5% | 3.21 | 3.11 |
| BOOST CS | 0.603 | 92.1% | 24.4 | **3.36** | 3.17 | 0.565 | 93.9% | **3.51** | **3.28** |
| BOOST Joint | 0.588 | 95.0% | 29.7 | 3.32 | **3.23** | 0.559 | 95.4% | 3.40 | 3.19 |
| *Setting:* Alpaca *few-shot* | | | | | | | | | |
| Base Model | 0.552 | 92.2% | 22.5 | 3.19 | 3.03 | 0.546 | 92.4% | 3.27 | 2.89 |
| Lex | 0.581 | 95.7% | 30.8 | 3.26 | 3.03 | 0.551 | 95.3% | 3.26 | 2.92 |
| BOOST CS | 0.608 | 94.6% | 28.4 | **3.38** | **3.18** | 0.584 | 94.8% | **3.40** | 3.10 |
| BOOST Joint | 0.591 | 95.7% | 30.1 | 3.36 | **3.18** | 0.572 | 95.2% | 3.37 | **3.14** |

Table 3: Intra-Group evaluation results on two benchmarks: CommonGen (with reference) and CSK-PN (without reference). Here, we **_define a group_** as multiple systems under the same setting (*i.e.,* base model) *and* on the same dataset. We use boldface to denote the best scores within each group, and underlines to denote the second best. Our model BOOST consistently achieves the most commonsensical ratings as annotated by humans. The gap between BOOST and the corresponding Base Model is statistically significant ($p<0.05$) measured by Student's t-test. Note that the human ratings across groups are *not* directly comparable as they are conducted in separate batches.

**GPT3/ChatGPT** We instruct OpenAI's `3.5-turbo` and `text-davinci-003` to generate a plausible sentence given the constraints, stating that the keywords do not necessarily have to remain in the same order. Note that these models are likely to be trained on our test data already.

For all compared systems, we decode with top_k ($k = 30$) sampling with a temperature $T = 0.7$.

### 4.4 Evaluation Setup

**Evaluation Metrics** We use the keyword coverage ratio (after lemmatization) and the $\mathcal{O}$ score as automatic metrics to assess the quality of generated texts. For the CommonGen benchmark which contains human written sentences, we also report the n-gram overlap (BLEU-4). Considering that our systems are trained towards higher $\mathcal{O}$ score, we also conduct human annotation for unbiased evaluation. Specifically, we instruct the MTurkers to evaluate 1) how commonsensical each sentence is from a 1-4 Likert scale, and 2) how much they

like the sentence overall (*e.g.,* being interesting and informative). An example questionnaire with the full instructions can be found in Appendix C. We pay the MTurkers $18 per hour, and the annotation process is the same as mentioned in §3.3.

**Inter-Group and Intra-Group Comparison.** Our human evaluation is relative, meaning that the human evaluators are asked to compare the quality of different machine-generated outputs given the same input constraint. Since we have five base models and each entails a group of systems to compare with, we first conduct human evaluation within each group. Then, we select representative systems for inter-group comparison.

## 5 Result and Analysis

### 5.1 Intra-Group Results

We compile the results on the CommonGen and CSK-PN benchmark in Table 3. We find out that,

| Test Data | CommonGen | | CSK-PN | |
| --- | --- | --- | --- | --- |
| **Human Eval** | **CS** | **Overall** | **CS** | **Overall** |
| A*esque | 3.07 | 2.81 | 3.09 | 3.04 |
| GeLaTo | 3.15 | 2.78 | / | / |
| Boost$_\text{gpt2}$ | 3.27 | 2.95 | 3.24 | 3.00 |
| Alpaca warm up | 3.27 | 3.05 | 3.21 | 3.01 |
| Alpaca few-shot | 3.32 | 3.20 | 3.20 | 3.10 |
| Boost$_\text{Alpaca warm up}$ | 3.40 | 3.17 | 3.41 | 3.16 |
| Boost$_\text{Alpaca few-shot}$ | 3.44 | **3.28** | 3.38 | **3.18** |
| Text-Davinci-003 | 3.33 | 3.19 | 3.33 | 3.10 |
| ChatGPT | **3.46** | 3.09 | **3.49** | 2.95 |
| Human | 3.49 | 2.99 | / | / |

Table 4: Inter-Group human ratings. The scores of all models are comparable within the same test benchmark. We color human performance in a grey background, and use boldface/underlines to denote the best/second-best scores among all machines.

BLEU-4 has a high correlation with the keyword coverage ratio ($r = 0.914$ measured by Pearson Correlation), but has close to zero correlation with human judgment on commonsense ($r = -0.08$) and overall preference ($r = 0.04$). We therefore hypothesize that BLEU-4, coverage ratio, and other metrics measuring the superficial lexical overlap with ground truth, cannot identify meaningful and commonsensical outputs at least in our setting.

Moreover, in all eight groups of human evaluation, Boost successfully improves the commonsense level and overall preference. Comparing Flan-T5 with gpt2, we see that our approach is more effective on instruction-tuned models than similarly-sized decoder only models. In addition, although Boost Joint achieves slightly lower commonsense ratings than Boost CS, the later is a lot worse in the keyword coverage, indicating that Boost CS has a higher risk to generate reasonable sentences without satisfying the input constraints. Hence, in the constrained generation setting, we still consider Boost Joint as the best model.

## 5.2 Inter-Group Results

The inter-group evaluation results are shown in Table 4. Our model Boost outperforms all baselines, including Davinci-003. We leave the comparison with ChatGPT in §6 as a separate discussion.

Surprisingly, although **human written references** are still the most commonsensical, they **are less preferred** by our annotators compared with Alpaca/Boost generations. Upon further inspection, we find out that the gold references in CommonGen are relatively short and flat (*e.g., "The car drove through the snow."*), which may also explain why Alpaca warmed up on CommonGen are

| Constraint | table, dog, game, walk, fireplace (from CommonGen) |
| --- | --- |
| Gelato | A dog is playing a game on a table next to a fireplace. |
| A* Decoding | A group of people are walking and playing video games at their dining room with fireplaces, tables, and dogs. |
| Davinci-003 | The dog walked around the table playing a game by the fireplace. |
| Boost Joint | The dog walked around the table while we played a game by the fireplace. |
| Reference | The dog plays the game of walking from the table to the fireplace. |
| Constraint | statue, liberty, alive (from CSK-PN) |
| A* Decoding | There are still some people who want to see statues of liberty as living creatures. |
| Alpaca | The Statue of Liberty became alive on a bright and sunny day. |
| Lex | The statue of Liberty is alive and stands proudly in New York City. |
| Davinci-003 | The Statue of Liberty stands alive and proud. |
| Boost Joint | The Statue of Liberty is a symbol of freedom and justice that is alive and well in the hearts of all Americans. |
| Constraint | ant, eat, telephone (from CSK-PN) |
| Lex | The ant was eating the phone as if it were a delicious snack. |
| Davinci-003 | The ant was seen eating a telephone. |
| Boost Joint | An ant eating a dead fly on the telephone. |
| Boost CS | A black ant eating on the side of a brown telephone. |

Table 5: Example generations by different systems. Full outputs of all compared models can be found in Table 7 in the Appendix.

less preferred than the few-shot setting where high-quality in-context examples are carefully selected.

## 5.3 Case Study

We show three example generations by our systems and the baselines in Table 5 to further understand the advantage of Boost. In the first example, the baselines connect different constraints logically, but in a less plausible way (e.g., all concepts are bonded to the same object). Our system on the other hand describes a scene where *people* play games while *dogs* walk around. In the second and third example, we all know that the Statue of Liberty is not alive and a telephone is inedible. Instead of directly adding negations, we observe Boost tends to provide more contexts to make its output reasonable. In contrast, other baselines wrongly acknowledge that the Statue of Liberty can be alive or the ant can eat a telephone.

## 6 Has ChatGPT solved this task?

**Pair-wise comparison with Boost.** According to Table 4, our Boost model may not surpass

| Winning System | BOOST CS | Same | ChatGPT |
|---|---|---|---|
| CS | 30% | 17% | 53% |
| Overall | 47% | 25% | 28% |

Table 6: Which system has better commonsense (CS) and overall human preference? Pair-wise comparison between BOOST and ChatGPT shows that our model earns more overall pick while the ChatGPT have higher commonsense.

ChatGPT in terms of commonsense, but it excels in overall preference. On the CSK-PN eval set where the gap between our model and ChatGPT is larger, we randomly select 100 pairs of outputs and conduct pairwise comparison on both commonsense and overall preference. Results can be found in Table 6. Specifically, each pair is first randomly shuffled and then annotated by at least two annotators. If the two annotators disagree, a third annotator is introduced for the final judge. They can also provide an optionally justification for their choice, which can earn them a small bonus.

Analysis of human's feedback reveal that ChatGPT tends to generate a sentence with highly common scenarios (*e.g., "It is not advisable to wear sunglasses at night as it can impede your vision and increase the risk of accidents."*), making the raters less interested. On the other hand, our model tends to provide more creative context (*e.g., "Someone wears sunglasses at night to avoid the bright lights of the approaching car."*), earning human annotators' overall preference without sacrificing the commonsense too much. As one annotator commented, *"I am fed up with those sentence with the so-called better commonsense because they are unimpressive"*. Such tendency of ChatGPT results in a higher commonsense rating yet noticeably lower overall preference. In short, we highlight that ChatGPT has not entirely solved the task.

**The (so far) impossible fair comparison.** Last, we would like to list two points regarding why evaluating ChatGPT and our model may not be a fair comparison: *(1) Test Data Contamination*: ChatGPT, which is trained on data up to 2021, likely have been trained on both datasets we tested on, including the test set. *(2) Size and Trick Differences*: Different from BOOST, ChatGPT is more than a plain language model and benefits largely from RLHF and many engineering tricks unknown to the public. It is also much larger than our largest PTLM, which is alpaca-7b. Nonetheless, our approach is technically complementary with Chat-

GPT's language model, too. Unfortunately, due to API limitations, direct verification remains infeasible as we do not have access to its output logits.

## 7 Related Works

**Controllable Generation with Frozen PTLMs** There are two major lines: modifying the decoding algorithm and guiding PTLMs with auxiliary models. Recently, Lu et al. (2021, 2022) propose neurologic decoding with a look ahead heuristic, and (Qin et al., 2022) propose energy-based constrained decoding. One drawback of this line that the inference is slow due to the large search space. In the other line, Dathathri et al.; Krause et al. (2021); Yang and Klein (2021) guide the generation process with an auxiliary model in a plug-and-play fashion by leveraging statistical principles such as the Bayesian rule. Meng et al. (2022) propose to solve the distributional discrepancy of training data and PTLM's generated tokens by training with data directly sampled from the base model. However, mistakes in commonsense are neglected when previous works formulate the whole task as a lexical constrained generation game.

**Commonsense Metrics** Zhou et al. (2022) measures the commonsense of dialogue turns by hard and soft matching the relations across each turn to ConceptNet. ACCENT (Ghazarian et al., 2023) propose an unsupervised metric to measure the event commonsense of dialogue responses via the ATOMIC knowledge graph (Hwang et al., 2020). Our commonsense oracle is inspired by ACCENT but we primarily focused on factoid commonsense in a constrained generation setting. A concurrent work of ours is Vera (Liu et al., 2023), a supervised model that learns to estimate the plausibility of statements. On the other hand, our metric is unsupervised and neuro-symbolic, thus more interpretable.

## 8 Conclusion

We present BOOST, a framework to boost the commonsense in PLTMs' generation by training an auxiliary model with a commonsense scorer as the oracle. Our $\mathcal{O}$-Scorer is task-agnostic and reference-free, meaning that it is generalizable to many downstream tasks such as dialogue and open-ended text generation. For such application, one may need to replace the vanilla PTLMs with task-specific models and then train the NADO head. The $\mathcal{O}$-Scorer can also be combined with task-specific guidance.

## Acknowledgement

We thank PlusLab members from UCLA and the anonymous EMNLP reviewers for their constructive feedback and suggestions that helped to improve the paper.

## Limitations

We discuss the limitations of our work. First, our tuple extractor covers only four relation types and can miss many other important relation types such as causal, temporal order, etc. These later relation types are more sophisticated such that LLMs are strong as gpt-3.5-turbo will fail at (Gao et al., 2023; Yuan et al., 2023; Chan et al., 2023; Bang et al., 2023). Second, we find out that the cosine similarities of sentence embeddings used in Eq. 1 to compute the compatibility scores sometimes do not align with human judgement. The errors incurred during the generative scoring process is then propagated into the training process of NADO, which negatively affect the output's quality. Last, although the auxiliary models have much smaller size than the PTLMs, the number of samples needed to train $R^{\mathcal{O}}$ is still large in order to guarantee a good approximation of the closed form solution derived in Eq. 4.

## Ethics Statement

It is known that the generated results by PTLMs could capture the bias reflected in the training data (Sheng et al., 2019; Wallace et al., 2019). Our model BOOST is build upon PTLMs including T5 (Raffel et al., 2020), GPT-2 (Radford et al., 2019), and Alpaca (Taori et al., 2023), which may potentially generate offensive content for certain groups or individuals. We suggest to carefully examine the potential biases before deploying the models to real-world applications.

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

# Appendix

## A    Few-Shot GPT-3.5 Tuple Extractor Prompt

The prompt we used to query gpt-3.5-turbo is displayed in Figure 4. Recall that we need the LLM to accurately extract both sensical tuples (e.g., a girl *is Capable Of* blowing candles) and nonsensical tuples (e.g., horse *is Capable Of* riding bikes) from the input sentence. Hence, not all sentences in the prompt are reasonable.

## B    Full Results of Case Study

The full results of the case study with outputs of all compared systems can be found in Table 7.

## C    Human Evaluation Questionnaire

Figure 5 and Figure 6 are screenshots of the questionnaire we used in the human evaluation.

| Constraint | table, dog, game, walk, fireplace (from CommonGen) |
|---|---|
| Gelato | A dog is playing a game on a table next to a fireplace. |
| A* Decoding | A group of people are walking and playing video games at their dining room with fireplaces, tables, and dogs. |
| Davinci-003 | The dog walked around the table playing a game by the fireplace. |
| Lex | A dog is walking around the fireplace and playing a game on the table. |
| Alpaca | The dog walks around the table while playing a game by the fireplace. |
| BOOST CS | Two dogs are playing a game of Frisbee by the fireplace next to the table. |
| BOOST Joint | The dog walked around the table while we played a game by the fireplace. |
| ChatGPT | A dog played near the table by the fireplace after a walk. |
| Reference | The dog plays the game of walking from the table to the fireplace. |

| Constraint | statue, liberty, alive (from CSK-PN) |
|---|---|
| A* Decoding | There are still some people who want to see statues of liberty as living creatures. |
| Davinci-003 | The Statue of Liberty stands alive and proud. |
| Lex | The statue of Liberty is alive and stands proudly in New York City. |
| Alpaca | The Statue of Liberty became alive on a bright and sunny day. |
| BOOST CS | The statue of liberty stands alone as a symbol of liberty and awakening alive. |
| BOOST Joint | The Statue of Liberty is a symbol of freedom and justice that is alive and well in the hearts of all Americans. |
| ChatGPT | The Statue of Liberty looked alive in the glowing sunset. |

| Constraint | ant, eat, telephone (from CSK-PN) |
|---|---|
| A* Decoding | A man is feeding ants to an antennae on top of his head, so they can be eating from the telephone. |
| Davinci-003 | The ant was seen eating a telephone. |
| Lex | The ant was eating the phone as if it were a delicious snack. |
| Alpaca | The ant ate the telephone. |
| BOOST CS | A black ant eating on the side of a brown telephone. |
| BOOST Joint | An ant eating a dead fly on the telephone. |
| BOOST CS | A black ant eating on the side of a brown telephone. |
| ChatGPT | The ant tried to eat the speaker of a miniature telephone. |

Table 7: Full results of case study by different systems.

<−− *Instruction :* −−>
Extract tuples (A, B) from the sentence for the relations based on the description below.
Do not infer anything. Only extract tuples that explicitly mentioned in the sentence.
Put None if there are no tuples to extract.

IsUsedFor: A (an object) is used to do B (a goal).
AtLocation: A is at the location or larger area B.
CapableOf: A (a living) is capable of doing B (an event)
PartOf: A is part of B.

<−− *Examples:* −−>
The runner ran because he wanted to win the car race.
IsUsedFor: None
AtLocation: None
CapableOf: (runner, run), (runner, win the car race)
PartOf: None

The small plates add dimension and depth to this dish of baked zucchinis and carrots.
IsUsedFor: (small plates, adding dimension to this dish)
AtLocation: None
CapableOf: None
PartOf: (baked zucchini, dish), (carrot, dish)

The grinning boy put his foot into the sock to dress himself.
IsUsedFor: None
AtLocation: (foot, sock)
CapableOf: (boy, grin), (grinning boy, put his foot into the sock)
PartOf: (foot, grinning boy)

The judges give high scores to the woman wearing a long dress who sings beautifully into a microphone on the
     stage.
IsUsedFor: (microphone, singing)
AtLocation: (woman, stage), (microphone, stage), (long dress, stage)
CapableOf: (judges, give high scores), (woman, wear a long dress), (woman, sing beautifully)
PartOf: None

A man is kicking a soccer ball with his head.
IsUsedFor: None
AtLocation: None
CapableOf: (man, kick a soccer ball with his head)
PartOf: (head, man)

I used a chisel and hammer to carve a piece of wood.
IsUsedFor: (chisel, carving a piece of wood), (hammer, carving a piece of wood)
AtLocation: None
CapableOf: (I, carve a piece of wood), (I, use chisel), (I, use hammer)
PartOf: None

Spewing volcano with waterfalls flowing results in an idyllic uncontaminated environment at summer in the
     mountains.
IsUsedFor: None
AtLocation: (volcano, mountains), (waterfalls, mountains)
CapableOf: (volcano, spew), (waterfall, flow)
PartOf: (waterfall, spewing volcano), (idyllic uncontaminated environment, mountains)

A fan argues with stewards after being told to leave the pitch.
IsUsedFor: None
AtLocation: (fan, pitch), (stewards, pitch)
CapableOf: (fan, argue with stewards), (stewards, tell fans to leave pitch)
PartOf: None

The soldier is driving the smiling tank across the bridge to save people.
IsUsedFor: (tank, driving)
AtLocation: (soldier, across the bridge), (tank, across the bridge)
CapableOf: (soldier, drive the tank), (tank, smile)
PartOf: None

Figure 4: Few-Shot prompt used to query gpt-3.5-turbo to extract tuples from a sentence. We purposefully select a few non-sensical sentences in the prompt.

**Task: commonsense annotation.** In this task, you will be given several sentences. **You are asked to rate how 'commonsensical' each sentence is** (namely, how well the sentence aligns with your commonsense knowledge of the world).

**We define 'commonsense knowledge' as the following:**

Commonsense knowledge includes the basic facts about events (including actions) and their effects, facts about knowledge and how it is obtained, facts about beliefs and desires. It also includes the basic facts about material objects and their properties.

**Rate each sentence from 1 to 4:**
- **4 means the most 'commonsensical'**; likely, you cannot find any issues that hurts commonsense
- **3 means mostly 'commonsensical' but a bit strange or has minor mistakes**; you probably can spot a minor point is weird, but would agree the sentence overall is still commonsensical
- **2 means somewhat 'commonsensical' with noticable mistakes or too general (i.e., would be helpful to add more context)**; for example, almost half of the sentence doesn't make too much sense to you
- **1 indicates the least 'commonsensical'.**; for example, most of the sentence doesn't make too much sense to you

**When you do not choose 4, you will be asked to indicate which word or phrase is hurting the commonsense. Please focus on the commonsesense part and ignore the minor grammatical mistakes if they exist.** After that, you will rate how you like the sentence overall, where you can consider various other aspects.

**Now let's take a look an example task that you are asked to complete.**

**Example 1**
**Sentence: Dog palying with frisbee in outdoors park on green grass.**

How commonsensical is this sentence?
○ **1. least commonsensical**    ○ **2. somewhat (but with noticable mistakes)**    ○ **3. mostly (can feel a bit strange)**    ⦿ **4. very commonsensical**

  **Which part of this sentence is not making sense:** N/A

Explanation: It aligns to our commonsense that dogs play with frisbee, and dogs love to go to outdoor park with green grass. Although this sentence has minor grammar errors, we would like to give it a score of 4.

**Example 2**
**I laughed so hard, I couldn't keep my foot sticking to my face.**

How commonsensical is this sentence?
○ **1. least commonsensical**    ○ **2. somewhat (but with noticable mistakes)**    ⦿ **3. mostly (can feel a bit strange)**    ○ **4. very commonsensical**

  **Which part of this sentence is not making sense:** keep my foot sticking to my face

Explanation: This sentence is strange because people don't always keep their foot sticking to their faces. Based on your personal feelings, we consider both scores of 2 or 3 as reasonable.

**Example 3**
**In supermarket the employee watched as the customer prepared their food to perfection.**

How commonsensical is this sentence?
○ **1. least commonsensical**    ⦿ **2. somewhat (but with noticable mistakes)**    ○ **3. mostly (can feel a bit strange)**    ○ **4. very commonsensical**

  **Which part of this sentence is not making sense:** customer prepared their food

Explanation: This sentence does not align well with our commonsense because in supermarkets, employees should perpare the food for the customer. However, we do not rate it

Figure 5: The instructions for human evaluation (page 1).

○ **1. least commonsensical**    ● **2. somewhat (but with noticable mistakes)**    ○ **3. mostly (can feel a bit strange)**    ○ **4. very commonsensical**

     **Which part of this sentence is not making sense:** customer prepared their food

Explanation: This sentence does not align well with our commonsense because in supermarkets, employees should perpare the food for the customer. However, we do not rate it as "least" commonsensical as it is still somewhat reasonable that because customer and employee are common concepts related to supermarkets.

---

**Example 4**
**A man is holding a tree and sitting on it.**

How commonsensical is this sentence?
● **1. least commonsensical**    ○ **2. somewhat (but with noticable mistakes)**    ○ **3. mostly (can feel a bit strange)**    ○ **4. very commonsensical**

     **Which part of this sentence is not making sense:** A man is holding a tree and sitting on it.

Explanation: First, trees grow from the ground and large, so that humans cannot hold them. Second, it is not possible to hold an object while sitting the same object.

---

**After you rate the commonsense score, we also want you to rate how much you like this sentence overall.** This part is pretty open-ended and have no correct answer, and you are encouraged to consider all aspects, including but not limited to coherence, funniness, creativeness, grammar, and the commonsensical score you previously rated.

---

**Hint: An efficient way to boost your HIT's approval rate -**

1) Read the sentences carefully. Each HIT consists of 2 groups of sentences. Each group consists of 5 sentences talking about the same topic. We kindly ask you to compare different sentences.

2) We have taken measures to prevent cheating and if you do not complete the task honestly we will know and the HIT will be rejected.

3) We keep records on the time you spend. So please make sure you have carefully read, and put enough thoughts on each word.

---

1. **Sentence 1: ${sentence_1}**

(a) How commonsensical is this sentence?
○ **1. least commonsensical**    ○ **2. somewhat (but with noticable mistakes)**    ○ **3. mostly (can feel a bit strange)**    ○ **4. very commonsensical**

     **Which part of this sentence is not making sense:** [                    ]

(b) How much do you like this sentence overall?
○ **1. little**    ○ **2. somewhat**    ○ **3. good**    ○ **4. best**

---

2. **Sentence 2: ${sentence_2}**

(a) How commonsensical is this sentence?
○ **1. least commonsensical**    ○ **2. somewhat (but with noticable mistakes)**    ○ **3. mostly (can feel a bit strange)**    ○ **4. very commonsensical**

     **Which part of this sentence is not making sense:** [                    ]

(b) How much do you like this sentence overall?
○ **1. little**    ○ **2. somewhat**    ○ **3. good**    ○ **4. best**

Figure 6: The instructions for human evaluation (page 2).