# OpenReview forum: "Harnessing Black-Box Control to Boost Commonsense in LM's Generation"
_EMNLP/2023/Conference — EMNLP 2023 Main_

### Official Review · Reviewer_bDkz · 2023-07-24

**Soundness:** 4

**Excitement:**

3: Ambivalent: It has merits (e.g., it reports state-of-the-art results, the idea is nice), but there are key weaknesses (e.g., it describes incremental work), and it can significantly benefit from another round of revision. However, I won't object to accepting it if my co-reviewers champion it.

**Paper Topic And Main Contributions:**

This paper aims to address the generative commonsense impotence problem of Pre-Trained Language Models (PLTMs). To this end, the authors propose a framework named BOOST, which first build a scorer to evaluate how commonsensical a sentence is, and then use the score to train an auxiliary model that steers the PTLM toward more commonsensical outputs. The contributions are three-fold:

1) The authors propose a reference-free scorer to evaluate how commonsensical a sentence is.
2) The authors extend the controllable generation method called NADO to the controllable commonsense generation field.
3) Extensive experiments validate the effectiveness of the proposed method.

**Questions For The Authors:**

Question A: In Table 2, it is shown that the Reference-Based matrix "BERTScore-all" is more consistent with human rating. Besides, I guess computing the BERTScore-all will be faster than the model inference. So I wonder why you choose a model-based method (i.e., train a T5-large) to serve as the Commonsense Scorer?

**Reasons To Accept:**

1. The paper is well-structed and easy to follow. The details of the proposed framework are also well illustrated.

2. The experiments are comprehensive and convincing. The authors evaluate their framework on a series of GPT-2-based and Alpaca-based language models on two constrained concept-to-sentence benchmarks. They also utilize various metrics to assess the performance.

**Reasons To Reject:**

1. The method lacks novelty to some extent. BOOST is composed of two main part, the first part is to construct a commonsense scorer by grounding the sentence to a dynamic commonsense knowledge base from four different relational aspects, the second part is to use the scorer as the oracle for a method called NADO [1], which has been proposed for controllable generation. Therefore, the method can be regarded as NADO applied to controllable commonsense generation, with the oracle is constructed based on the commonsense domain knowledge.

[1] Meng, T., Lu, S., Peng, N., & Chang, K. W. (2022). Controllable text generation with neurally-decomposed oracle. Advances in Neural Information Processing Systems, 35, 28125-28139.

**Reproducibility:**

4: Could mostly reproduce the results, but there may be some variation because of sample variance or minor variations in their interpretation of the protocol or method.

**Reviewer Confidence:**

3: Pretty sure, but there's a chance I missed something. Although I have a good feel for this area in general, I did not carefully check the paper's details, e.g., the math, experimental design, or novelty.

---

> ### Author Rebuttal · Authors · 2023-08-28
>
> Dear reviewer, thank you so much for your valuable review! We would like to address your questions and concerns as follows:
>
> **[QA].  In Table 2, it is shown that the Reference-Based matrix "BERTScore-all" is more consistent with human rating. Besides, I guess computing the BERTScore-all will be faster than the model inference. So I wonder why you choose a model-based method (i.e., train a T5-large) to serve as the Commonsense Scorer?**
>
> - It is infeasible to use BERTScore-all as the Commonsense Score exclusively, simply because of its reference-based nature. In our settings, 1) all data used to train the NADO head, which are self-sampled by PTLMs, and 2) some of the eval data (i.e., CSK-PN dataset) do not come with a reference. In light of these, we have to train a T5 model to serve as the reference-free Commonsense Scorer. After that, BERTScore-all serves as a compared system when test our newly proposed reference-free Scorer rigorously – on a small subset of eval data which is actually paired with ground truth.
>
> **[Contribution/Novelty/Significance of Work]**
>
> - First, we would like to kindly reiterate that proposing a novel framework for controllable generation is not one of our claimed contributions. Rather, we are the first to extend NADO from a simple, binary-rule based setting to a continuous, neural-evaluator based setting and show its efficacy. To this end, we also propose a new reference-free commonsense evaluator that can be used as an off-the-shelf tool. To the best of our knowledge, the idea of using automatic eval metric to guide language models towards more commonsense-compliant outputs is novel and ``hasn't been explored before``.
>
> - Moreover, commonsense-compliance is a known challenge among LMs. Many downstream fine-tuned models such as open-domain chatbots and story writing systems would largely benefit from improved commonsense abilities. Our proposed framework stands out for its model-agnostic nature. Added to the task-agnostic and reference-free nature of our O-Scorer, 	``Boost can be easily applied to benefit various downstream tasks for better commonsense-aligned generation``.
>
> With the above, we sincerely hope you could kindly reconsider our merits. Thank you!

---

### Official Review · Reviewer_DEwH · 2023-08-02

**Typos Grammar Style And Presentation Improvements:** NA
**Soundness:** 4

**Excitement:**

4: Strong: This paper deepens the understanding of some phenomenon or lowers the barriers to an existing research direction.

**Missing References:**

There are some missing references in lines 578-582 to evidence the statements.

Gao, Jinglong, et al. "Is ChatGPT a Good Causal Reasoner? A Comprehensive Evaluation." arXiv preprint arXiv:2305.07375 (2023).

Yuan, Chenhan, Qianqian Xie, and Sophia Ananiadou. "Zero-shot temporal relation extraction with chatgpt." arXiv preprint arXiv:2304.05454 (2023).

Chan, Chunkit, et al. "Chatgpt evaluation on sentence level relations: A focus on temporal, causal, and discourse relations." arXiv preprint arXiv:2304.14827 (2023).

Bang, Yejin, et al. "A multitask, multilingual, multimodal evaluation of chatgpt on reasoning, hallucination, and interactivity." arXiv preprint arXiv:2302.04023 (2023).

**Paper Topic And Main Contributions:**

This paper proposes a reference-free commonsense metric to assess how commonsensical a sentence is, first extracting tuples from a sentence and then grounding the extracted tuples to a dynamic CSKB (i.e., COMET). Moreover, the paper extends a controllable generation approach to improve commonsense for black-box PTLM by presenting a framework to boost the commonsense in PLTMs’ generation through training an auxiliary model with a commonsense scorer as the oracle.

**Questions For The Authors:**

Question A: Lack of some instruction-tuned baselines for comparison (e.g., text-davinci-003 and flan-t5), instead of instruction tuning again on gpt2 model. text-davinci-003 api (as well as flan-t5) can return the output distribution and fulfill the requirement stated in line 400.

Question B: ChatGPT whether solving this task remains a question and hasn’t been included in this paper. Try to include this evaluation to validate the ability of ChatGPT to generates a plausible sentence given a list of concepts as input and show the gap between ChatGPT and the suggested method in this paper.

Question C:  Whether these generated commonsense outputs can be applied to benefit any downstream tasks to emphasize the significance of this task (such as the open-domain dialogue task).

**Reasons To Accept:**

* This paper conducts an interesting task by generating a plausible sentence given a list of concepts as input.
* The performance of this proposed method outperforms baselines
* This paper proposes a novel reference-free commonsense metric to assess how commonsensical a sentence is.

**Reasons To Reject:**

* Lack of some instruction-tuned baselines for comparison (e.g., text-davinci-003 and flan-t5), instead of instruction tuning again on gpt2 model.
* Moreover, ChatGPT whether solving this task, which generates a plausible sentence given a list of concepts as input, remains a question and hasn’t been included in this paper.
* Whether these generated commonsense outputs can be applied to benefit any downstream tasks to emphasize the significance of this task (such as the open-domain dialogue task stated in [1])

[1] Sarik Ghazarian, Yijia Shao, Rujun Han, Aram Galstyan, and Nanyun Peng. 2023. Accent: An automatic event commonsense evaluation metric for open-domain dialogue systems.

**Reproducibility:**

4: Could mostly reproduce the results, but there may be some variation because of sample variance or minor variations in their interpretation of the protocol or method.

**Reviewer Confidence:**

3: Pretty sure, but there's a chance I missed something. Although I have a good feel for this area in general, I did not carefully check the paper's details, e.g., the math, experimental design, or novelty.

---

> ### Author Rebuttal · Authors · 2023-08-28
>
> Dear reviewer, thank you so much for your valuable review and constructive feedback! We address your questions and comments as follows:
>
> **[Question A]. Lack of some instruction-tuned baselines for comparison (e.g., text-davinci-003 and flan-t5), instead of instruction tuning again on gpt2 model. text-davinci-003 api (as well as flan-t5) can return the output distribution and fulfill the requirement stated in line 400.**
>
> Thank you for your constructive suggestion! We agree it would be nice to test our approach on instruction-tuned models, too. To our knowledge, however, we couldn't access the output logits of text-davinci-003 through the OpenAI API; we only have access to the top 5 tokens per step. Hence, we quickly implemented our approach with Flan-T5. Under this setting, the NADO head is a 4-layer t5 decoder that takes in the input constraints as hidden steps from a fixed, pretrained encoder. Due to time constraints, we are only able to test our approach on zero-shot Flan-T5-large (780M), and conduct automatic evaluation.
>
> Please find the automatic evaluation results in the following table.
>
> | Test   Data       | CommonGen       |          |       | CSK-PN          |          |
> |-------------------|-----------------|----------|-------|-----------------|----------|
> |       Metric      | CS Score - Mean | Coverage | BLEU4 | CS Score - Mean | Coverage |
> | flan_t5 zero-shot |                 |          |       |                 |          |
> |    Base Model    |      0.571      |   84.6%  |  17.5 |      0.555      |   80.7%  |
> |        Lex        |      0.577      |   93.7%  |   26.0  |      0.569      |   89.6%  |
> |    Boost CS   |      0.619      |   91.3%  |  21.6 |      0.613      |   88.9%  |
> |   Boost Joint  |      0.606      |   93.1%  |  25.6 |      0.601      |   89.6%  |
>
> Table 1: Automatic eval results. We implement the same algorithm on Flan-T5.
>
> **Analysis**: Comparing the above results with gpt2-large with warm up (774M, in Table 3 in the original paper), we see that our approach is actually more effective on instruction-tuned models than similarly-sized decoder only models.
>
> ---
> \
> **[Question B]. ChatGPT whether solving this task remains a question and hasn’t been included in this paper. Try to include this evaluation to validate the ability of ChatGPT to generates a plausible sentence given a list of concepts as input and show the gap between ChatGPT and the suggested method in this paper.**
>
> First, in response to including the comparison with ChatGPT, we'd like to address several points regarding why evaluating ChatGPT and our model may not be a fair comparison:
>
> - **Test Data Inclusion**: ChatGPT, which is trained on data up to 2021, likely have been trained on both datasets we tested on (CommonGen is published in 2020 while CSK-PN is a subset of ConceptNet which is published in 2017), including the test set.
>
> - **Size and Trick Differences**: Different from ours, ChatGPT is more than a plain language model and benefits largely from RLHF and many unknown engineering tricks. It is also much larger than our largest PTLM, which is alpaca-7b.
>
> **Experimental Results**: Upon request, we still obtained the results with ChatGPT below. Following previous decoding settings, we use sampling with a temperature T=0.7.
> - **Result 1: Human evaluation results across multiple baselines.**
>
> |Human Eval| CommonGen - CS |CommonGen - Overall | CSK-PN - CS |  CSK-PN - Overall|
> | -------- | ------- |  ------- |------- |  ------- |
> |A*esque| 3.07 |2.81| 3.09 |3.04|
> |GeLaTo| 3.15| 2.78 |/ |/|
> |BOOST gpt2| 3.27| 2.95| 3.24| 3.00|
> | -------- | ------- |  ------- |------- |  ------- |
> |Alpaca warm up| 3.27| 3.05| 3.21 |3.01|
> |Alpaca few-shot| 3.32 |[3.20]| 3.20| 3.10|
> |BOOST Alpaca warm up| [3.40] |3.17| **3.41**| [3.16]|
> |BOOST Alpaca few-shot| **3.44** |**3.28**| [3.38] |**3.18**|
> | -------- | ------- |  ------- |------- |  ------- |
> |Text-Davinci-003| 3.33 |3.19| 3.33| 3.10|
> |ChatGPT| 3.46 |3.09| **3.49**| 2.95|
> |Human |**3.49** |2.99| /| /|
>
> Table 2: Evaluation results with ChatGPT. We use boldface to denote the best scores within each group, and brackets [] to denote the second best.
>
> - **Result 2: Pairwise comparison**
>
> Furthermore, on the CSK-PN eval set where the gap between our model and ChatGPT is larger, we randomly select 100 samples and then conduct pairwise comparison on both commonsense and overall quality. Specifically, each pair is first randomly shuffled and then annotated by at least two Turkers. If the two annotators disagree, a third person is introduced for the final judge.
>
>
> **(a) Which system has better commonsense?**
>
> |Boost CS|Same |ChatGPT|
> | -------- | ------- |  ------- |
> |30%|17%|53%|
>
> Table 3 (a). Pairwise comparison between Boost CS and ChatGPT about commonsense.
>
>
> **(b) Which system has better overall quality?**
>
> |Boost CS|Same |ChatGPT|
> | -------- | ------- |  ------- |
> |47%|25%|28%|
>
> Table 3 (b). Pairwise comparison between Boost CS and ChatGPT about overall quality.
>
> From both Table 2 and Table 3, we see that our model BOOST has better overall quality than ChatGPT, but slightly worse in commonsense. Upon further inspection, we notice that on the CSK-PN eval set that contains concepts linked with negated commonsense relations (e.g.``<wear sunglasses, at night>``), ChatGPT tends to generate a short sentence with negation (e.g., ``It is ridiculous to wear sunglasses at night.``). Such tendency of ChatGPT results in a higher commonsense rating yet noticeably lower overall preference. On the other hand, our model tends to provide more context to make its output reasonable (e.g., ``Someone wears sunglasses at night to avoid the bright lights of the approaching car.``), earning human annotators’ overall preferences without sacrificing the commonsense too much.
>
> **Conclusion**: In short, it's crucial to highlight that ChatGPT hasn't entirely "solved" the problem, and has much room for improvement. Last, we want to emphasize that ``our approach can be complementary with ChatGPT’s language model, too``. Unfortunately, due to API limitations, direct verification remains infeasible as we do not have access to its output logits.
>
> ---
> \
> **[Question C]. Whether these generated commonsense outputs can be applied to benefit any downstream tasks to emphasize the significance of this task (such as the open-domain dialogue task).**
>
> Yes! Our O-Scorer is task-agnostic and reference-free, meaning that our framework is generalizable to many downstream tasks including but not limited to dialogue and open-ended text generation. For such application, we just need to replace the vanilla PTLMs with task-specific models and then train the NADO head. In addition, the O-Scorer can also be combined with task-specific guidance such as the lexical checking function in our task. We shall add related discussions into our paper to emphasize the significance of this work.
>
> **Missing reference**: Thank you, we shall add these to our paper!
>
> With the above, we sincerely hope you could kindly reconsider our merits and experimental soundness. Thank you!

---

### Official Review · Reviewer_Kqgt · 2023-08-05

**Typos Grammar Style And Presentation Improvements:** In line 216, "PTL)" -> "PTLM"
**Soundness:** 3

**Excitement:**

3: Ambivalent: It has merits (e.g., it reports state-of-the-art results, the idea is nice), but there are key weaknesses (e.g., it describes incremental work), and it can significantly benefit from another round of revision. However, I won't object to accepting it if my co-reviewers champion it.

**Paper Topic And Main Contributions:**

To generate more commonsensial text based on large language model (LLM), this paper proposes a reference-free evaluation method O-score for checking the commonsense knowledge in the generated text, and utilizes the O-score to guide the controllable generation method NADO by training a neural network-based model to approximate the O-score. Experiments are conducted on two constrained concept-to-sentence benchmarks. The contribution can be summarized:
1. this paper proposes a reference-free evaluator to assess how commonsensial a setence is.
2. this paper extends a controllable generation method to improve commonsense for pretrained language model.

**Questions For The Authors:**

1. how to train the additional NADO head?
2. since O-score is a reference-free evaluator, then what does O(x,y) in Eq.(2) mean?
3. what does y_i mean in Eq.(3)?
4. In Table 3, on the CommonGen, BOOST CS achieves the majority of SOTA according to O-score and Human CS & OVERALL metrics, but its BLEU4 score only performs better than base model, how to explain the phenomenon?
5. In Table 5, why are the baseline models listed in the three examples inconsistent?

**Reasons To Accept:**

1. A new evaluation method O-score is proposed to check the commonsense for the generated text.
2. Based on NADO, a concept-to-sentence model is proposed to improve the commonsense by considering the O-score.

**Reasons To Reject:**

1. Some details are not clear enough, especially the section 2.3.
2. The experimental results are not convicing.

**Reproducibility:**

2: Would be hard pressed to reproduce the results. The contribution depends on data that are simply not available outside the author's institution or consortium; not enough details are provided.

**Reviewer Confidence:**

3: Pretty sure, but there's a chance I missed something. Although I have a good feel for this area in general, I did not carefully check the paper's details, e.g., the math, experimental design, or novelty.

---

> ### Author Rebuttal · Authors · 2023-08-26
>
> Dear reviewer, thank you for your valuable review! We’ve noticed that there might have been some misunderstandings, and we believe that most of your questions and clarity concerns are to be adequately addressed in our forthcoming rebuttal. We sincerely hope for your re-evaluation of our merits and strengths based on the responses.
>
> We carefully address your concerns and questions as follows:
>
> - **[Q1]. how to train the additional NADO head?**
>
> You can find the details of the training method (input, output, notation, formulation, loss, etc.) in Section 2.3.2 line 256-284. You can also find the training data which is generated by self-sampling  in Section 4.1 line 372-382, and auxiliary model architecture in Section 4.2 line 418-427.
>
> Please kindly let us know if you have more specific questions. If so, we are more than happy to address them in detail during the author-reviewer discussion period.
>
> - **[Q2]. since O-score is a reference-free evaluator, then what does O(x,y) in Eq.(2) mean?**
>
> In Eq 2 and the entire paper, the $x$ is an input constraint (e.g., ‘lasso, horse, cow’) and $y$ is a  ``machine-generated sentence`` that follows the input constraints (e.g., A horse is being lassoed by a cow.), and O(x,y) means the score assigned to the input-output pair (x, y). Namely, **_O-score is indeed reference-free_** because it does not require any ground-truth sentences to judge the commonsense. If we only care about the commonsense but not lexical constraint, O(x,y) is reduced to O(y). We keep using O(x,y) in Eq.(2) because in our experiments, we also combine the lexical checking Boolean function Lex(x,y) with the O-score.
>
> You may find the official notations (e.g., meaning of x and y) in Section 2.1 line 110-124.
>
> - **[Q3]. what does y_i mean in Eq.(3)?**
>
> This is a typo, and it should be $y_{T'}$ which means $y_{=T'}$. In Eq.(3) we haven’t introduced what $i$ is.
> We sincerely apologize for such a typo in an important equation and _**thank you so much**_ for spotting it/pointing it out!!
>
> - **[Q4]. In Table 3, on the CommonGen, BOOST CS achieves the majority of SOTA according to O-score and Human CS & OVERALL metrics, but its BLEU4 score only performs better than base model, how to explain the phenomenon?**
>
> Thank you for your insightful question! We do have a few hypotheses to explain this phenomenon which we unfortunately weren't able to touch upon in the submission due to the page limit. The short answer is that ``BLEU-4, which was originally introduced for machine translation, is not a good automatic metric for our task focusing on commonsense and overall text quality (such as being meaningful and informative)``.
>
> **First**, for example, you can see in Table 3 that the keyword Coverage ratio has an exceptionally _**high correlation**_ with BLEU-4 (``0.914`` measured by Pearson Correlation and ``0.785`` measured by Spearman Correlation). This indicates that in our constrained generation task, BLEU-4 is biased towards favoring machine outputs that merely cover all the input concepts, ignoring aspects of commonsense. **On the other hand**, we also find the references in the CommonGen dataset mediocre. As is stated in Line 509-513, we find out that the references in CommonGen are relatively short and flat (e.g., The car drove through the snow.), which may also explain why Alpaca warmed up on CommonGen are less preferred by human annotators than few-shot Alpaca. In fact, BLEU-4 has close to zero correlation with human judgment on commonsense (``-0.08`` measured by Pearson; ``0.11`` measured by Spearman) and overall quality (``0.04`` measured by Pearson; ``-0.05`` measured by Spearman).
>
> Based on the above, we hypothesize that BLEU-4, or any other metric that measures the superficial lexical overlap with ground truth, cannot differentiate high-quality and commonsensical outputs in our settings.
>
> - **[Q5]. In Table 5, why are the baseline models listed in the three examples inconsistent?**
>
> Apologies for the confusion. ``Due to page limit, we select the better/more informative baselines for each case, and that’s why they are different``. Namely, we have to be selective only to fit in the page limit. An exception goes to Gelato (and references) as we do not have the model’s output (or ground truth) on the CSK-PN eval set.
>
> In case you are curious, we compile the full results below:
> | **Constraint**    | **table, dog, game, walk, fireplace (from CommonGen)** |
> | -------- | ------- |
> | Gelato  | A dog is playing a game on a table next to a fireplace.    |
> | A* Decoding | A group of people are walking and playing video games at their dining room with fireplaces, tables, and dogs.     |
> | Lex    | A dog is walking around the fireplace and playing a game on the table.   |
> | Base Model | The dog walks around the table while playing a game by the fireplace.|
> | BOOST CS| Two dogs are playing a game of Frisbee by the fireplace next to the table.|
> | BOOST Joint| The dog walked around the table while we played a game by the fireplace.|
> | Davinci-003 | The dog walked around the table playing a game by the fireplace.|
> | Reference| The dog plays the game of walking from the table to the fireplace.|
> |ChatGPT| A dog played near the table by the fireplace after a walk.|
>
>
>
> | **Constraint**    | **statue, liberty, alive (from CSK-PN)**|
> | -------- | ------- |
> | A* Decoding | There are still some people who want to see statues of liberty as living creatures.   |
> | Lex    | The statue of Liberty is alive and stands proudly in New York City.  |
> | Base Model | The Statue of Liberty became alive on a bright and sunny day.|
> | BOOST CS|  The statue of liberty stands alone as a symbol of liberty and awakening alive.|
> | BOOST Joint| The Statue of Liberty is a symbol of freedom and justice that is alive and well in the hearts of all Americans.|
> | Davinci-003 | The Statue of Liberty stands alive and proud.|
> | ChatGPT| The Statue of Liberty looked alive in the glowing sunset.|
>
>
> | **Constraint**    | **ant, eat, telephone (from CSK-PN)**|
> | -------- | ------- |
> | A* Decoding |   A man is feeding ants to an antennae on top of his head, so they can be eating from the telephone.|
> | Lex    | The ant was eating the phone as if it were a delicious snack.  |
> | Base Model | The ant ate the telephone.|
> | BOOST CS|  A black ant eating on the side of a brown telephone.|
> | BOOST Joint| An ant eating a dead fly on the telephone.|
> | Davinci-003 | The ant was seen eating a telephone.|
> | ChatGPT| The ant tried to eat the speaker of a miniature telephone.|
>
>
>
>
> - **6. Reproducibility:** We will open-source all codes upon acceptance.
>
>
> In conclusion, we sincerely hope that the above response helps to clarify your questions and concerns. With the inclusion of an extra page upon acceptance, we shall be able to incorporate more details to adequately address all your comments. In the camera ready version, we also intend to present the methodology section (especially Section 2.3 talking about NADO) and the experimental results more comprehensively. _**Finally, we hope you agree that these adjustments are fixable by revisions of writings instead of another round of submission. We sincerely hope you could kindly reconsider our merits and contributions.**_ Thank you!

---

### Meta-Review · Area_Chair_NAPd · 2023-09-17

**Recommendation:** 4

**Metareview:**

The paper proposes a novel approach to improve the commonsensical output of pre-trained language models (PLTMs). The research focuses on (1) the creation of a reference-free evaluator called the O-score to assess the commonsense nature of generated text and (2) guides the extension of a controllable generation method named NADO, aiming to steer PLTMs towards producing more commonsensical text.

Pros:
1. Introduces a new metric, the O-score, to evaluate the commonsensicality of generated text without requiring reference sentences.
2. Extends the NADO method to focus on controllable commonsense generation. The proposed method is model-agnostic.
3. The paper is well-structured and comprehensive in its experimentation, and the results surpass baselines.

Cons:
1. The initial manuscript misses out on comparing with more diverse and strong LLMs such as instruction-tuned LLMs. However, I can see these experiments are added during rebuttal. I hope the author will add these new results in the final version and provide more details discussions. For example, the reply to QA of Reviewer DEwh should have more narrative to explain the results and comparisons.
2. The paper does not clarify how the generated commonsense outputs could be useful for downstream applications, which could attest to the method's significance. The related discussion in rebuttal should be included in the final version.

---

### Decision · Program_Chairs · 2023-10-07

**Decision:**

Accept-Main

**Comment:**

The paper proposes a novel approach to improve the commonsensical output of pre-trained language models (PLTMs). The research focuses on (1) the creation of a reference-free evaluator called the O-score to assess the commonsense nature of generated text and (2) guides the extension of a controllable generation method named NADO, aiming to steer PLTMs towards producing more commonsensical text.

Pros:
1. Introduces a new metric, the O-score, to evaluate the commonsensicality of generated text without requiring reference sentences.
2. Extends the NADO method to focus on controllable commonsense generation. The proposed method is model-agnostic.
3. The paper is well-structured and comprehensive in its experimentation, and the results surpass baselines.

Cons:
1. The initial manuscript misses out on comparing with more diverse and strong LLMs such as instruction-tuned LLMs. However, I can see these experiments are added during rebuttal. I hope the author will add these new results in the final version and provide more details discussions. For example, the reply to QA of Reviewer DEwh should have more narrative to explain the results and comparisons.
2. The paper does not clarify how the generated commonsense outputs could be useful for downstream applications, which could attest to the method's significance. The related discussion in rebuttal should be included in the final version.